



# Global evolution of flux transfer events along the magnetopause from the dayside to the far tail

Yann Pfau-Kempf[1], Konstantinos Papadakis[1], Markku Alho[1], Markus Battarbee[1], Giulia Cozzani[1,2], Lauri Pänkäläinen[1], Urs Ganse[1], Fasil Kebede[1,3], Jonas Suni[1], Konstantinos Horaites[1], Maxime Grandin[4,1], and Minna Palmroth[1,4]

[1]Department of Physics, University of Helsinki, Helsinki, Finland
[2]LPC2E, CNRS/CNES/University of Orléans, Orléans, France
[3]Department of Physics and Technology, University of Bergen, Bergen, Norway
[4]Finnish Meteorological Institute, Helsinki, Finland

**Correspondence:** Yann Pfau-Kempf (yann.kempf@helsinki.fi)

**Abstract.** Magnetic flux ropes are structures of magnetic field rolled-up along a longitudinal axis, which are forming in a variety of magnetised plasmas. In near-Earth space, flux ropes are a manifestation of energy transfer at the magnetopause and in the magnetotail current sheet. We present a new method to detect magnetic flux ropes in large-scale simulations, using only magnetic field line tracing. The method does not require prior identification of structures of interest such as current sheets or
null lines, and thus allows one to identify flux ropes of any size and orientation, anywhere in the simulation domain. In this work, the new method is implemented in the hybrid-Vlasov model Vlasiator and demonstrated in global simulations of the terrestrial magnetosphere.

We study the evolution of flux ropes forming during flux transfer events on the dayside magnetopause under southward interplanetary magnetic field. It is found that flux ropes with an axial orientation along the dawn-dusk direction and propagating
beyond the cusps will rapidly reconnect with the lobe magnetic field and vanish. In contrast, the flux ropes remaining near the equatorial plane and with an axial orientation along the flow direction, that is tangential to the magnetopause, can maintain their structure and propagate tens of Earth radii down the tail in the absence of a reconnecting shear magnetic field component. These results are a step forward in the global characterisation of flux ropes in and around the magnetosphere, and may help in guiding the search for elusive far-tail flux ropes in satellite measurements.

**1 Introduction**

Magnetic flux ropes are structures characterised by axially oriented magnetic field around which twisting magnetic field is wrapped with an increasing angle with respect to the axial direction, reminiscent of fibres twisted in a rope. They have been observed or inferred in a variety of plasma environments especially when magnetic reconnection occurs. They form in the Sun, pierce its surface where they can erupt (e.g., Wang et al., 2017; MacTaggart et al., 2021), and propagate in the solar
wind as smaller-scale flux ropes with scale sizes of the order of $10^5$ km (e.g., Moldwin et al., 2000; Cartwright and Moldwin, 2010) or large magnetic clouds or interplanetary coronal mass ejections spanning $10^7$ km and more (e.g., Janvier et al., 2014).





In the near-Earth environment, dynamic magnetopause reconnection leads to the formation of flux ropes called flux transfer events (FTE), which have been studied extensively ever since their first in situ detection (Haerendel et al., 1978; Russell and Elphic, 1978; Rijnbeek and Cowley, 1984), up to and including large statistical surveys using spacecraft constellations (e.g.,
Wang et al., 2006; Lv et al., 2016; Kieokaew et al., 2021). When the magnetotail current sheet disrupts and reconnects, flux ropes form that are usually called plasmoids (see e.g., Eastwood and Kiehas, 2015, for a review), which can also nowadays be studied statistically thanks to extensive observational datasets (e.g., Smith et al., 2024). They also form in the ionospheres of unmagnetised planets and in the magnetospheres of magnetised planets, and have been observed at Mercury (e.g., Slavin et al., 2009; Sun et al., 2016; Zhong et al., 2023), Venus (e.g., Elphic and Russell, 1983; Zhang et al., 2012), Mars (e.g., Brain
et al., 2010; Hara et al., 2017a, b; Bowers et al., 2021), Jupiter (e.g., Kronberg et al., 2005; Vogt et al., 2014; Sarkango et al., 2021, 2022) and its moon Ganymede (Romanelli et al., 2022), Saturn and Titan (e.g., Jackman et al., 2014; Jasinski et al., 2016; Martin et al., 2020), Uranus (DiBraccio and Gershman, 2019, who also note a lack of observations of Neptunian flux ropes so far), as well as near comets (Edberg et al., 2016). Flux ropes have also been assumed to form in astrophysical contexts such as black hole accretion disks (e.g., Ripperda et al., 2022). Flux ropes are therefore quite a fundamental and universal phenomenon
in magnetised space plasmas.

As noted above, depending on the context flux ropes may be called flux transfer events when considering magnetopause reconnection, or plasmoids in magnetotail or other planetary contexts. Some authors distinguish flux ropes with a strong axial field from plasmoids without axial field, thus with a more cylindrical rather than helical geometry, which can form under very symmetric conditions. In the case of two-dimensional simulations, the term 'magnetic island' is sometimes encountered (e.g.,
Fermo et al., 2012; McGregor et al., 2013; Zhou et al., 2014; Pfau-Kempf et al., 2016; Hoilijoki et al., 2019). In this work we use the term 'flux rope' to cover all such rolled-up magnetic field structures forming as part of magnetic reconnection.

Considering in situ observational data, the classical signature of a flux rope passing by the spacecraft is a bipolar oscillation of the magnetic field component $B_N$ normal to both the axis of the flux rope and the direction of propagation of the flux rope. It is usually seen in the time series of magnetic field components, optionally after transforming into a coordinate system
maximising the variance of the oscillating component. If the spacecraft does not cross close to the axis yet there is the suspicion of a passing flux rope nearby, assumptions can be made about the properties of the flux rope and a fit to a model equation can be made to solve for flux rope orientation and size. This is the principle of the Grad-Shafranov reconstruction or more recent methods (see, e.g., Isavnin et al., 2011, for a review). A further example is the method developed by Huang et al. (2018) to detect FTEs by correlating observed magnetic field signatures with a characteristic 'target function to be correlated' built
upon an idealised, cylindrical flux rope configuration. This is essentially a refinement of the identification of the bipolar $B_N$ signature.

When considering simulation data analysis, ad hoc tracing of field lines is commonly used to show the existence of flux ropes. Paul et al. (2022) developed an automated method to detect and track FTEs in their global magnetospheric simulation output. They first identify FTEs by inspecting $B_N$ signatures on the magnetopause. They then use an algorithm that builds
a tree representation of the data cube of thermal pressure in one simulation snapshot. Each local thermal pressure maximum and the surrounding volume are assigned an index that allows tracking in consecutive simulation snapshots. The obtained





dendrogram is then 'pruned to get rid of other high-pressure regions in the domain that are not of interest' and identified high-pressure structures are matched to the FTEs. This method thus allows to assign a connected simulation volume characterised by a thermal pressure signature to a given FTE.

In the context of remote-sensing data such as the observation of flux ropes at the Sun's surface, in the corona and beyond, techniques have been developed taking advantage of magnetograms of the solar surface as well as optical observations (e.g., Isavnin et al., 2014; Liu, 2020; Wagner et al., 2024). Derived methods can be applied to detect flux ropes in simulations of the Sun's surface and corona. For example Lowder and Yeates (2017) calculate the magnetic field lines' helicity and the twisted magnetic field of flux ropes is characterised by high helicity, peaking at the centre of the flux rope. This is used to define

thresholds that allow to define the flux rope footpoints in the photosphere and the volume of the flux ropes over the surface.

  In this work, we introduce a new method to detect flux ropes in a general way in global simulations, without prior assumptions about their shape, orientation, or location near current sheets or other pre-determined structures of interest (Section 2). Our method can be applied to any magnetised plasma simulation setup in principle, and it is used in this work to detect and follow flux ropes comprehensively in a global, three-dimensional simulation of the Earth's magnetosphere performed with

the hybrid-Vlasov model Vlasiator (Palmroth et al., 2018; Ganse et al., 2023; Pfau-Kempf et al., 2024). The implementation is discussed in Section 2.2 and the global simulation setup is presented in Section 3. The results are presented in Section 4, allowing to characterise the output of the proposed flux rope identification algorithm and to track flux ropes comprehensively throughout the simulation run. In particular, the propagation of FTEs from the dayside to the far magnetotail flanks is studied. A discussion of the obtained results and the flux rope detection algorithm's parameters is given in Section 5 and we present our

conclusions in Section 6.

## 2 Flux rope identification algorithm

### 2.1 Method

The impetus for developing a new algorithm to detect flux ropes in a magnetospheric simulation comes from several realisations. Firstly, in simulations, flux ropes such as FTEs forming at and propagating along the magnetopause, or plasmoids

forming in magnetotail reconnection, can exhibit cross-sections ranging from the numerical grid scale at formation to several Earth radii ($1\,R_\mathrm{E} = 6371\,\mathrm{km}$). Secondly, their orientation can be arbitrary. Thirdly, the method should be independent from having to first identify interfaces such as the magnetopause or tail current sheets so as to identify flux ropes anywhere in the simulation volume. This leads to the requirements that the method should not rely on 1) arbitrary length scales such as a fixed absolute search radius or field line tracing length, 2) specific directions or coordinates, nor 3) prior identification of features

like current sheets or processes like magnetic reconnection. Furthermore, the variety of scales leads to the conclusion that a local method using only variables and their derivatives defined at a given simulation point is not sufficient to capture large and potentially complex flux rope configurations. This is discussed in more detail in Section 5.2.

  Since the most general feature of flux ropes is the twisting of their magnetic field $\boldsymbol{B}$, the algorithm is designed to determine where magnetic field lines are tightly wound. This is done by tracing the magnetic field backward and forward from every





(a) The seed point **is** in a flux rope.   (b) The seed point **is not** in a flux rope.

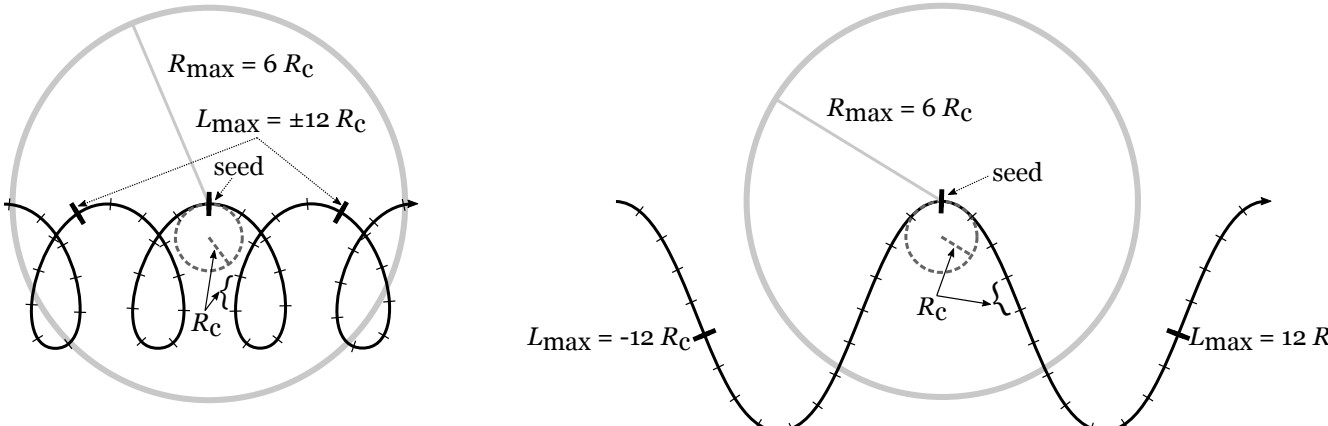

**Figure 1.** Illustration of the flux rope identification method for the parameters $L_\mathrm{max} = 12\,R_\mathrm{c}$ and $R_\mathrm{max} = 6\,R_\mathrm{c}$. Magnetic field lines are black. (a) The magnetic field is sufficiently twisted, when tracing the field for a length of $L_\mathrm{max}$ in either direction the radius $R_\mathrm{max}$ is not exceeded. This seed point is thus part of a flux rope. (b) The magnetic field is not very twisted, when tracing the field for a length of $L_\mathrm{max}$ in either direction the radius $R_\mathrm{max}$ is exceeded. This seed point is not part of a flux rope.

numerical grid point in the simulation up to a maximum tracing distance along the field line $L_\mathrm{max}$. If both the forward and backward parts of the field line did not exit a sphere with a radius of $R_\mathrm{max}$ centred on the seed point, the seed point is considered to be part of a flux rope. $L_\mathrm{max}$ and $R_\mathrm{max}$ are expressed in units of the curvature radius $R_\mathrm{c} = 1/|\boldsymbol{b} \cdot \nabla \boldsymbol{b}|$ (where $\boldsymbol{b} = \boldsymbol{B}/|\boldsymbol{B}|$), in order for the algorithm to adapt to the local scale of magnetic field structures. This method is illustrated in a cartoon fashion in Figure 1.

## 2.2 Implementation

The algorithm described above is implemented and optimised for runtime execution during full three-dimensional simulations of the Earth's magnetosphere using the hybrid-Vlasov model Vlasiator (Palmroth et al., 2018; Pfau-Kempf et al., 2024). Magnetic field tracing is performed using a simple, adaptive Euler algorithm (e.g., Press et al., 2011), limited to step lengths ranging 100–1000 km or 0.1–1 times the highest-resolution numerical grid cells used in magnetospheric simulation setups (see Section 3). The magnetic field is split into a static, curl-free background component and a propagated, perturbed component (von Alfthan et al., 2014; Palmroth et al., 2018). All variables related to the field solver are stored on a uniform, Cartesian grid at the finest resolution, and tracing is performed on this grid (Papadakis et al., 2022). During tracing, the background field components are obtained at arbitrary coordinates $(x, y, z)$ using the same analytic expressions that are used to set them at initialisation. The perturbed components are reconstructed to second order at the same coordinates $(x, y, z)$ using the formalism by Balsara (2009) used in Vlasiator's field solver as well (Palmroth et al., 2018). Reconstructing the field at any $(x, y, z)$ during tracing yields more accurate results than only using the components stored at grid locations. The reconstruction requires a comprehensive set of derivative values to be available. Storing them all in order to perform the tracing post hoc would require





tens of gigabytes of disk space per output file corresponding to a given time in the simulation, which is prohibitive. Hence the decision was made to perform this algorithm as part of in-situ data analysis, that is at runtime.

The seed points $(x_0, y_0, z_0)$ for tracing are taken as the centres of the cells of the spatially refined mesh which is used to store and propagate the plasma's velocity distribution function (Papadakis et al., 2022; Ganse et al., 2023; Kotipalo et al., 2024). At each seed point, the curvature radius $R_\mathrm{c}$ is determined. Then tracing is performed along $\pm\boldsymbol{B}$, recording the maximum extent $R^\pm = \max\left(|(x_0, y_0, z_0) - (x_n, y_n, z_n)|\right)$ over all successive $n$ tracing steps, until the tracing reaches the distance $L_\mathrm{max}$ along the field line. If both $R^+ < R_\mathrm{max}$ and $R^- < R_\mathrm{max}$ (like in Figure 1(a)), the value of $R_\mathrm{cutoff} = \max(R^+, R^-)$ is recorded for

the location $(x_0, y_0, z_0)$. If $R^+ > R_\mathrm{max}$ or $R^- > R_\mathrm{max}$ (like in Figure 1(b)), if a boundary of the domain is reached, or if tracing reaches values of $R^\pm$ significantly larger than the domain size, tracing is stopped (see Section 2.3). For the particular simulation run presented in this work, $L_\mathrm{max} = 12\,R_\mathrm{c} \approx 4\pi\,R_\mathrm{c}$, hence tracing is allowed to proceed up to almost two turns around an ideal, cylindrical configuration of radius $R_\mathrm{c}$. The maximum extent is set to $R_\mathrm{max} = 10\,R_\mathrm{c}$, thus values of $R_\mathrm{cutoff}$ in the range 0–10 are stored for each seed point, allowing for later determination of a suitable threshold to be used for analysis

(see Section 4.2).

## 2.3    Termination conditions

The flux rope detection algorithm is executed during large-scale simulation runs performed on tens of thousands of supercomputer cores. The simulation volume is decomposed into thousands of spatial domains mapped to individual computational tasks using the Message-Passing Interface (MPI). Field lines are traced from every grid cell through potentially large swathes of the

simulation volume. The implementation of the algorithm was optimised in terms of memory and inter-task communications in the context of the specific grid libraries and data structures used by the code. Further optimisations informed by the nature of the physical problem at hand are critical to avoid spending a large fraction of the computation time on this analysis. They are described here.

Other data products generated at runtime include tracing the magnetic field in the whole domain to obtain connection

information, allowing to determine the open and closed field regions. The flux rope detection is performed alongside the full-domain tracing, avoiding tracing the same field lines twice. That tracing naturally includes termination conditions at the inner and outer boundaries.

Due to the inaccuracy inherent to the discretisation of the problem, and especially in regions of tightly-wound field configurations near the numerical grid resolution, it is possible that the field line tracing circles around certain regions for long

distances without ever exiting the domain. Furthermore, during prototyping of the algorithm a peculiar structure was identified in the initialisation phase of the magnetospheric setup. When the magnetotail forms, a pair of magnetic field line loops forms at either edge of the tail current sheet, near the transition between the tail current sheet and the magnetosheath. These field lines stretch for tens of $R_\mathrm{E}$ along the tailward flow, that is the $x$-direction, turning around at the tips of this long and thin structure. In the $y$- and $z$-direction this structure is only a few $R_\mathrm{E}$ in size. In the middle of this structure, $R_\mathrm{c}$ is much larger than the

simulation domain, leading to a very large $L_\mathrm{max}$. While algorithmically correct, the detection of this type of structure is not relevant in the context of magnetospheric simulations, where flux ropes are considered to be bundles of magnetic flux winding



around their axial direction. Therefore, a termination condition interrupts the field line tracing if the traced distance $L^{\pm}$ reaches a limit. The limit defaults to the sum of the simulation domain size in every coordinate direction but it can also be set ad hoc by the user.

Finally, both the full-domain tracing and the flux-rope detection feature a parameter allowing to leave a fraction of cells unresolved. Once that limit is reached, the algorithm stops, providing another way to tune its computational cost.

## 3   Global magnetospheric simulation setup

Vlasiator is a hybrid-Vlasov simulation code modelling ions (only protons herein) using their discretised velocity distribution function while electrons are a charge-neutralising fluid. It is mainly tailored towards large-scale simulations of the terrestrial

magnetosphere and its surrounding magnetosheath–bow shock–foreshock system (von Alfthan et al., 2014; Palmroth et al., 2018). The code is openly available under the GNU GPL-2 license (Pfau-Kempf et al., 2024) and the model is typically run on hundreds of nodes on top-tier supercomputers due to the large memory and computational requirements (Ganse et al., 2023; Kotipalo et al., 2024).

     In this work, we present a simulation run in a volume spanning $[-110;50]\,R_{\mathrm{E}}$ in the $x$-direction and $[-58;58]\,R_{\mathrm{E}}$ in the

155 $y$- and $z$-directions of the Geocentric Solar-Magnetospheric (GSM) coordinate system. The base grid of $128 \times 92 \times 92$ cells yields a coarsest spatial resolution of $\Delta x = 8000\,\mathrm{km} = 1.26\,R_{\mathrm{E}}$ and it is statically refined up to three levels, yielding a finest spatial resolution of $\Delta x = 1000\,\mathrm{km} = 0.16\,R_{\mathrm{E}}$ in the tail current sheet and at the dayside magnetopause (Papadakis et al., 2022; Ganse et al., 2023; Kotipalo et al., 2024). The velocity space resolution is $\Delta v = 40\,\mathrm{km\,s^{-1}}$. The phase-space density threshold, below which the velocity distribution is neither stored nor propagated (von Alfthan et al., 2014; Palmroth et al.,

2018), is set to $10^{-15}\,\mathrm{m^{-6}s^3}$ where the proton number density is higher than $10^5\,\mathrm{m^{-3}}$, to $10^{-17}\,\mathrm{m^{-6}s^3}$ where the proton number density is lower than $10^4\,\mathrm{m^{-3}}$, and linearly interpolated in between. The $+x$ inflow wall maintains constant inflow and interplanetary magnetic field (IMF), the other walls maintain Neumann (copy) conditions ensuring outflow of plasma. The spherical inner boundary is centred around the origin of the simulation domain where the Earth is located and it is set at a radius of $4.7\,R_{\mathrm{E}}$. It couples the hybrid-Vlasov domain with an ionospheric model solving for the ionospheric potential using a

height-integrated conductivity model (Ganse et al., 2024). Field-aligned currents computed near the inner boundary are mapped along the magnetic field to the ionospheric grid. Using parameterised particle precipitation and a model atmospheric profile from the NRLMSISE model (Picone et al., 2002), ionisation rates and thus conductivities are computed and height-integrated, so that the electric potential can be solved for on the ionospheric grid. The gradient of the electric potential is mapped back into the hybrid-Vlasov domain and used to determine an electric field $\boldsymbol{E}$ and hence an $\boldsymbol{E} \times \boldsymbol{B}$ drift velocity that is given to the

ion velocity distribution functions near the inner boundary.

     The initial and inflow solar wind conditions are uniform and steady with a proton number density of $10^6\,\mathrm{m^{-3}}$, a temperature of $0.5\,\mathrm{MK}$ and a velocity of $(-750,0,0)\,\mathrm{km\,s^{-1}}$. For the initial simulation state, inside a radius of $15.7\,R_{\mathrm{E}}$, the velocity is gradually tapering from the solar wind velocity to zero at the inner boundary. The initial magnetic field is the unscaled,





unperturbed geomagnetic dipole with $0°$ tilt angle, gradually transitioning to the constant IMF of $(0,0,-5)\,\mathrm{nT}$ towards the
$+x$-direction.

## 4   Results

With its fast solar wind and moderate, purely southward IMF, the simulated setup produces active dayside reconnection. This generates FTEs as previously studied in 2D (Pfau-Kempf et al., 2016; Jarvinen et al., 2018; Hoilijoki et al., 2019; Akhavan-Tafti et al., 2020; Pfau-Kempf et al., 2020; Grandin et al., 2020; Ala-Lahti et al., 2022) and 3D (Pfau-Kempf et al., 2020;
Tesema et al., 2024; Grandin et al., 2024), which loads the magnetotail lobes and leads to reconnection of the tail current sheet. That in turn generates plasmoids and flux ropes as studied previously in 2D (Palmroth et al., 2017; Runov et al., 2021) and in 3D (Palmroth et al., 2023; Grandin et al., 2023). In Sections 4.1 and 4.2 we first illustrate how the method detects flux ropes throughout the simulation domain and how the outcome is affected by the choice of $R_{\mathrm{cutoff}}$. Sections 4.3 and 4.4 focus on describing the evolution of FTEs along the dayside and nightside magnetopause, respectively.

### 4.1   Global mapping of flux ropes in the magnetosphere

Figure 2 illustrates how flux ropes can be mapped in the magnetospheric simulation domain. The view is from the solar wind's direction towards Earth, that is parallel to the $-x$-direction. The grey surface of the region with closed field lines gives a proxy for the position of the dayside magnetopause. The algorithm described in Section 2 yields values of 0 when no flux rope is detected, and non-zero values up to $R_{\mathrm{max}}$ in case of detection. Points of detection up to $R_{\mathrm{cutoff}}$ values of 3, 5, and 7 are shown
in panels (a), (b), and (c), respectively, as spherical markers. To confirm the nature of the detections, magnetic field line stubs forward and backward from each detection point are plotted too. They are capped at a field line length of $L_{\mathrm{max}} = 12R_{\mathrm{c}}$ from the detection point. The colour of the spheres and lines shows the curvature radius at the seed point, given in units of $R_{\mathrm{E}}$. Figure 3 shows the same plotted information as viewed from North along the $-z$-direction.

     In both Figures 2 and 3 it is clear that structures with curvature radii of the order of $R_{\mathrm{c}} \approx 6\,R_{\mathrm{E}}$, coloured purple, are
inconsistent with the scale of flux transfer events on the dayside magnetopause, which itself has a curvature of a similar order of magnitude. In the same way in Figure 3 some structures with high $R_{\mathrm{c}}$ at the detection point are visible in the magnetotail. These detections are the result of the large curvature radius at the starting point, leading to the field line stretching out for tens of $R_{\mathrm{E}}$ but still less than $R_{\mathrm{max}}$. It can thus be useful to filter out curvature radii too large compared to the scale of the system under consideration, such as the magnetopause, but they are retained here for illustration purposes.

### 4.2   Sensitivity of $R_{\mathrm{cutoff}}$

Given that a single production-scale run with Vlasiator costs tens of millions of core-hours on modern supercomputers, the value of $R_{\mathrm{max}} = 10\,R_{\mathrm{c}}$ was chosen conservatively high. Figures 2 and 3 illustrate the effect of setting $R_{\mathrm{cutoff}}$ to the values of 3, 5, and 7 $R_{\mathrm{c}}$ (panels (a), (b), and (c), respectively).

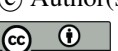

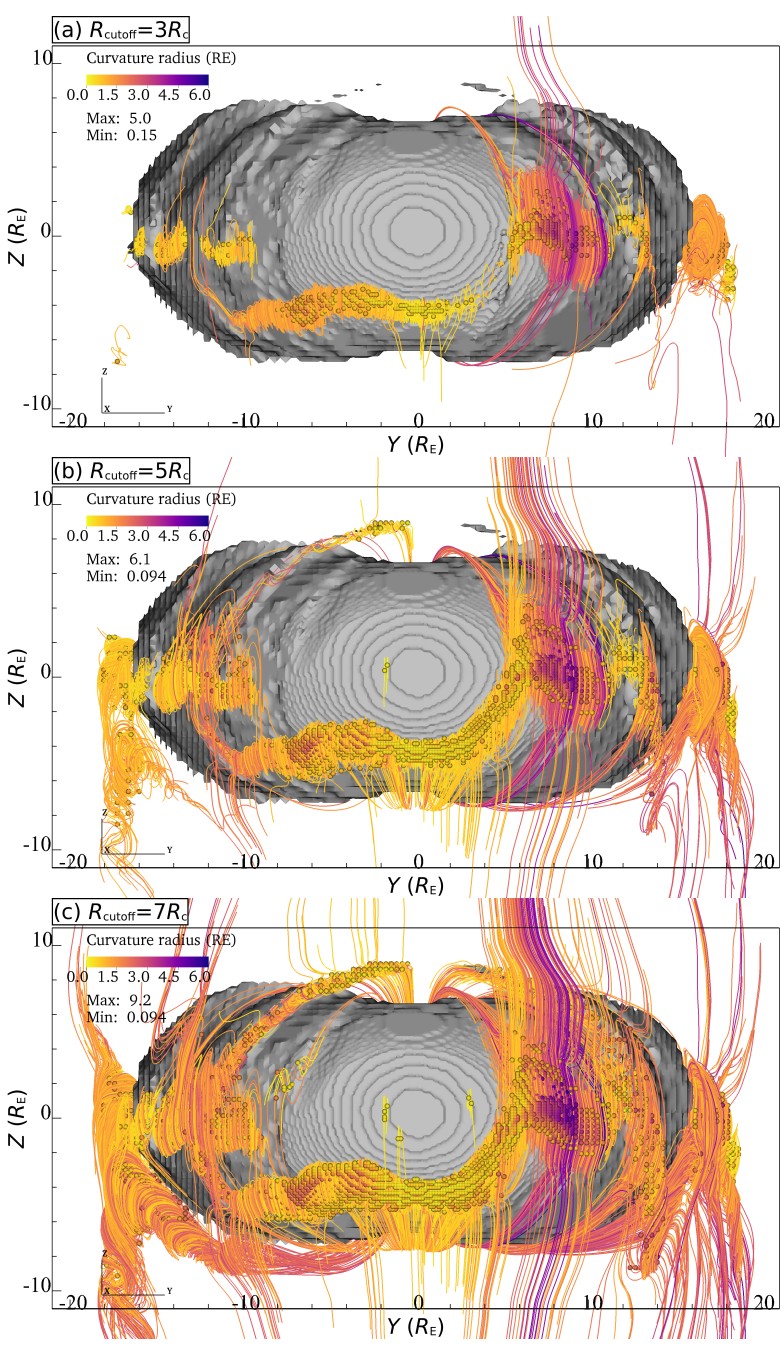

**Figure 2.** Flux ropes detected in the simulation at t=1600 s. View of the dayside from the direction of the solar wind, axes (GSM coordinates) in $R_E$. Grey: surface of the region of closed magnetic field lines connected at both ends to the inner boundary. Spheres: points near flux ropes detected by the algorithm. Lines: magnetic field lines traced from the detected points out to $R_{cutoff}$. Colour scale: curvature radius $R_c$ at detection point, in units of $R_E$. Panels (a)–(c): $R_{cutoff} = 3, 5, 7 R_c$.

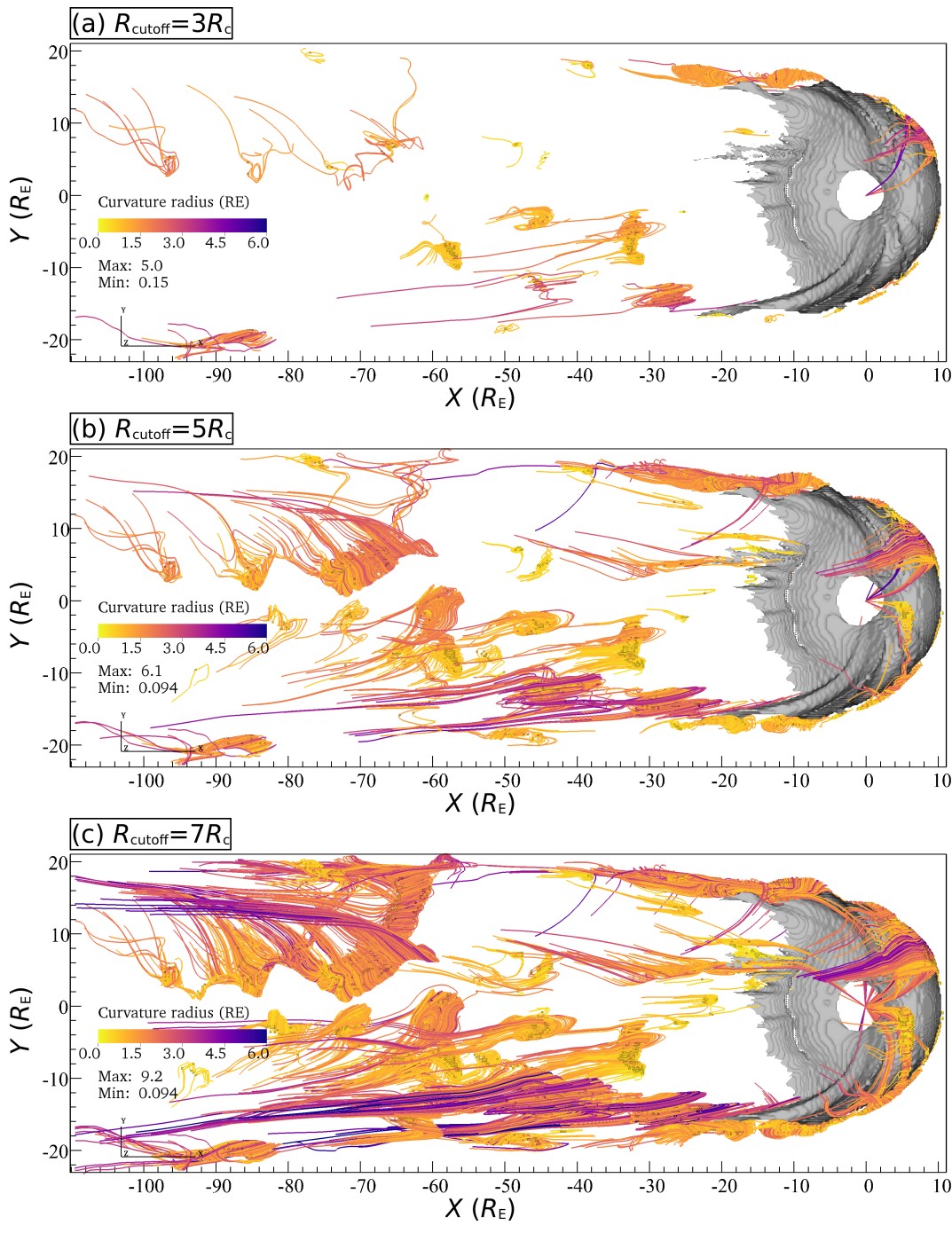

**Figure 3.** Same format as Figure 2, view from above the equatorial plane, showing wound-up magnetotail structures.





Comparing the panels in Figure 2, it is clear that too low an $R_\mathrm{cutoff}$ value can lead to failing to register flux ropes that are detected at higher values. Two prominent examples are the long flux rope in the $(-y,+z)$-quadrant and the curved flux rope in the $(-y,-z)$-quadrant, which are absent in panel 2(a) at $R_\mathrm{cutoff} = 3\,R_\mathrm{c}$, partially detected in panel 2(b) at $R_\mathrm{cutoff} = 5\,R_\mathrm{c}$ but well covered in panel 2(c) at $R_\mathrm{cutoff} = 7\,R_\mathrm{c}$. The same behaviour can be identified by comparing the flux ropes detected at the various levels of $R_\mathrm{cutoff}$ in the magnetotail as shown in Figure 3.

At even higher values of $R_\mathrm{cutoff}$ (not shown), especially for large curvature radii, the algorithm detects structures that fulfil the field line extent criterion but are not rolled up into flux ropes. They are generally bent field structures such as the magnetopause or tail current sheets.

The analysis of flux ropes therefore requires a careful choice of $R_\mathrm{cutoff}$. It has to be low enough to minimise the amount of false positives on the one hand, and high enough to detect also the more loosely wound structures on the other hand. Motivated by this analysis, a value of $R_\mathrm{cutoff} = 7\,R_\mathrm{c}$ is chosen for rest of this work.

## 4.3 Evolution of FTEs on the dayside

Under southward IMF conditions as in the simulation used in the present work, reconnection occurs at low latitudes on the dayside magnetopause (e.g., Trattner et al., 2021). Owing to the spatial and temporal variability of magnetic reconnection, FTEs form and are pushed along the magnetopause by the reconnection exhausts as well as the ambient magnetosheath plasma flow. We first investigate the evolution of the FTEs produced under these conditions in the subsolar region of the dayside magnetopause.

In Figure 4, the panels 4(a)–4(d) show the North-South velocity component $V_z$ in the $x-z$ plane at coordinates $y = -4.5, -1.5, 1.5, 4.5\,R_\mathrm{E}$ in colour, with the thin black magnetic field line stubs illustrating the general magnetic topology. As a further guide, the magnetopause is detected with the modified plasma $\beta$ parameter which includes the dynamic pressure, $\beta^* = 0.5$ (Xu et al., 2016; Brenner et al., 2021), and shown as a thick black contour. The purple X and yellow square markers denote X- and O-points, respectively, as detected with the method of Alho et al. (2024) based on the minimum gradient analysis (MGA) and minimum directional derivative (MDD) techniques. The regions detected as being near flux ropes at the level of $R_\mathrm{cutoff} = 7\,R_\mathrm{c}$ are marked in green. Naturally, detected flux ropes (green circles) are coinciding with O-points (yellow squares). The panels 4(e)–(f) show the latitude – magnetic local time (MLT) map of the open-closed magnetic field boundary (OCB) as a black contour in the North and South ionosphere (set at an altitude of $100\,\mathrm{km}$, as explained by Ganse et al., 2024), respectively. Additionally, the footpoints of flux ropes magnetically connected to the ionosphere are plotted on these maps as well. They are marked with a circle if the source point in the flux rope is at a coordinate $|y| < 4.5\,R_\mathrm{E}$ encompassed by the planes of panels (a)–(d), and a + marker otherwise. The footpoints are coloured according to the $(x,z)$-coordinate of their source point, following the two-dimensional colour map shown. This Figure 4 is a snapshot at simulation time $t = 1612\,\mathrm{s}$ from Supplementary Material Animation 1, which covers the time interval 1073–1612 s.

At numerous times and locations along the magnetopause, strong exhaust channels visible in $V_z$ on either side of X-lines confirm the occurrence of active reconnection, in a patchy and bursty fashion as investigated previously (Pfau-Kempf et al., 2020). A prominent reconnection exhaust channel can be seen for example in Figure 4(a)–(d) northward of the X-line at

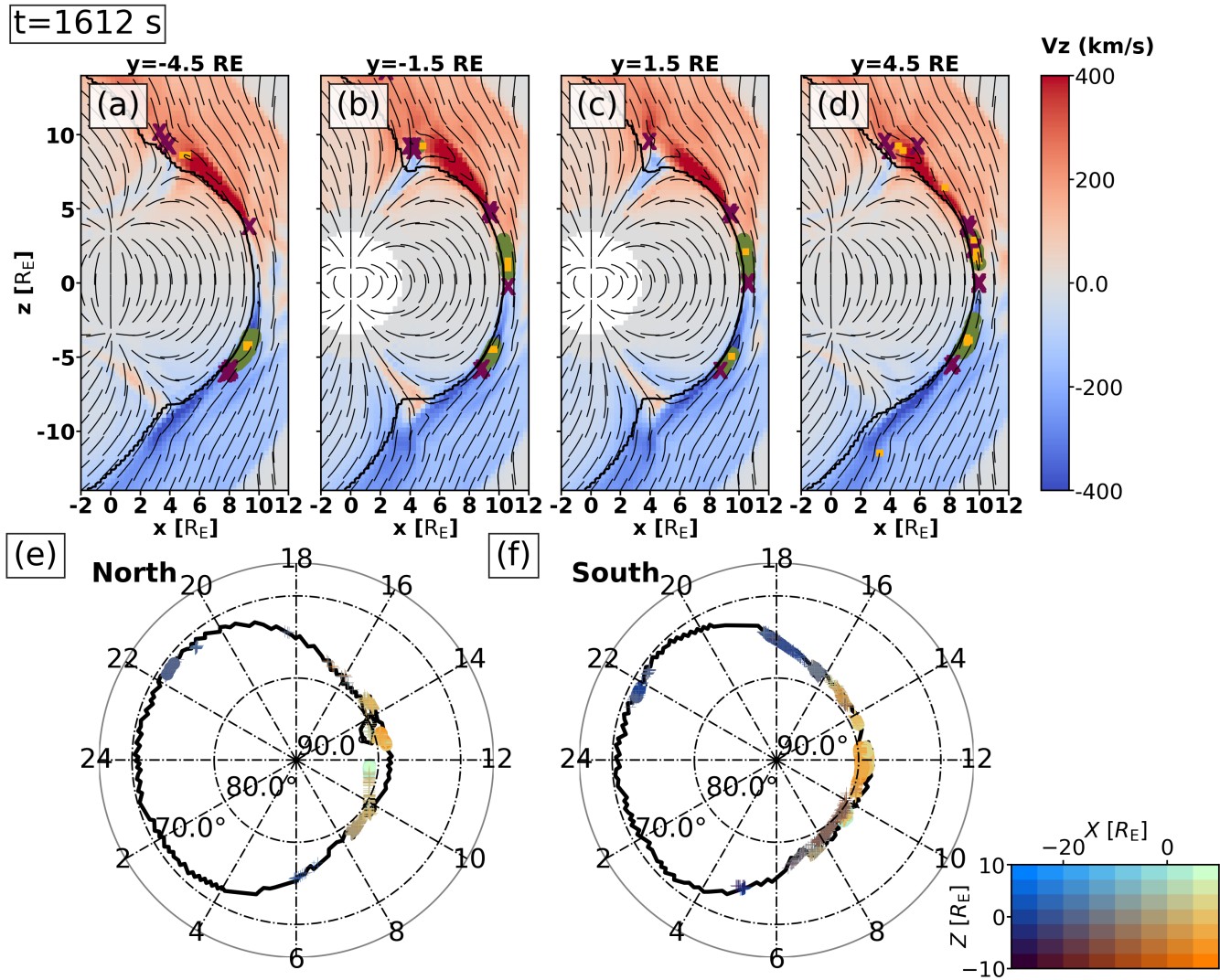

**Figure 4.** Panels (a)–(d): North-South velocity component $V_z$ (colour) in the $(x, z)$-plane at coordinates $y = -4.5, -1.5, 1.5, 4.5\,R_\mathrm{E}$. The thin black lines are tangent to the magnetic field. The thick black contour is set at $\beta^* = 0.5$ (Xu et al., 2016; Brenner et al., 2021) to show the magnetopause position. The purple X and yellow square markers denote X- and O-lines using the Alho et al. (2024) method. The green circles denote the regions where flux ropes are detected at the level of $R_\mathrm{cutoff} = 7\,R_\mathrm{c}$. Panels (e)–(f): North and South hemisphere ionospheric latitude – magnetic local time (MLT) map of the open-closed magnetic field boundary (black contour). The footpoints of detected flux ropes are marked with a circle if the source point in the flux rope is at a coordinate $|y| < 4.5\,R_\mathrm{E}$, and a + marker otherwise. The footpoints are coloured according to the $(x, z)$-coordinates of their source point, following the two-dimensional colour map shown. This figure is at $t = 1612\,\mathrm{s}$ from the beginning of the simulation. See the animated version of this figure for the time interval 1073–1612 s as Supplementary Material Animation 1.





$z = 4\,R_{\mathrm{E}}$. Many more FTEs are seen in Supplementary Material Animation 1 forming at lower $|z|$ and moving along the diverging magnetosheath flow northwards and southwards of the equatorial plane ($z = 0$).

Once the FTEs have reached the tailward portion of the cusp, their leading side comprises a magnetic field component antiparallel to the magnetic field of the magnetosphere, thus conducive to magnetic reconnection. At $t = 1612\,\mathrm{s}$ in Figure 4(a)–4(d), active reconnection is seen near $(x = 4\,R_{\mathrm{E}}, z = 9\,R_{\mathrm{E}})$. The remnants of the FTE are still denoted by the yellow squares and green circles, and an X-line is seen on the leading side. Enhanced flows are evidence of active reconnection exhaust channels, into the cusp (blue enhancement in panels 4(b) and 4(c)) as well as outwards (red enhancement northward of the X-

line in panel 4(c) in particular). Further clear examples of magnetic reconnection can be identified in Supplementary Material Animation 1 for example at the North cusp during the approximate times 1430–1450, 1530 and 1612 s. Some aspects of this FTE-cusp reconnection process have been studied using a magnetohydrodynamic model by Paul et al. (2023).

By definition, magnetic reconnection modifies the topological configuration of spatially adjacent magnetic domains and FTEs carry away newly-opened flux from the magnetosphere. It is thus expected to see signatures of FTEs in the OCB plotted

in Supplementary Material Animation 1 and Figure 4(e)–4(f). As a baseline, it can be observed that during the period of about 1150–1220 s, the OCB is mostly smooth and convex on the dayside in the absence of large FTEs perturbing the magnetopause. A number of FTEs form after 1200 s, their footpoints straddling the OCB, but they do not modify the position of the OCB significantly. The fine-scale jaggedness of the OCB is due to the triangular tessellation of the Fibonacci grid used for the ionosphere solver (Ganse et al., 2024). During the time interval 1350–1450 s, large FTEs travel both North and South towards

higher latitudes on the magnetopause, and they indent the OCB mostly between 9 and 15 MLT. A period with clearly identified indentations of the OCB is around 1415 s, when the OCB is indented at 11–14 MLT in the North and at 9–13 MLT in the South. Similar perturbations of the OCB are registered again from 1570 s to the end of the simulation, as is also visible in Figure 4(e)– 4(f). The footpoints show that the highest-latitude FTEs are connected the deepest in the open field region, poleward of the OCB in the dayside region, and the footpoints vanish at the same time as their source FTEs vanish through reconnection in the

cusps.

### 4.4   Evolution of FTE flux ropes on the nightside

Inspecting the outcome of the flux rope detection method with plots like Figure 3 and animations thereof (not shown) reveals that flux ropes are not limited to the dayside as presented in Section 4.3, nor to the tail current sheet (as studied with Vlasiator by Palmroth et al., 2023; Alho et al., 2024). There is a population of flux ropes forming at low $|z|$ on the dayside which travel

with the magnetosheath flow along the flanks of the magnetopause from the dayside to tens of $R_{\mathrm{E}}$ downtail on the nightside, as suggested by the detection of flank O-lines by Alho et al. (2024). Indeed Figure 4(e)–(f) as well as Supplementary Animation 1 show a significant population of flux rope footpoints that are connected to the morning or evening sector of the OCB. They are mostly denoted with + markers as their source flux rope is beyond the span of the plotted planes of panels 4(a)–(d), that is at $|y| > 4.5\,R_{\mathrm{E}}$. Furthermore, their colour indicates that they exist at low $|z|$, meaning they exist near the equatorial region

of the magnetopause. They lose their connection to the ionosphere eventually, as the footpoints are seen to disappear, but the flux ropes do not cease to exist, as shown in the following. A third population of flux rope footpoints can be discerned in





Figure 4(e)–(f) and Supplementary Animation 1 in the 21–3 MLT night sector. They are the signatures of plasmoids forming in magnetotail current sheet reconnection processes, as studied previously (Palmroth et al., 2023; Alho et al., 2024). Those will be the subject of separate studies and are thus left out of the scope of this work.

Figure 5 and its animated version Supplementary Material Animation 2 present a flux rope which is still relatively close to Earth around $x = -10\,R_{\mathrm{E}}$. It is the curved flux rope that is also visible in the $(-y, -z)$-quadrant of Figure 2(c). The morphology of the flux rope is described by the spatial cuts through the flux rope of panels 5(a) and 5(b) in two orthogonal planes. The in-plane components of the magnetic field are shown with the black tangent lines whereas the colour in the background shows the out-of-plane component. The green contour delimits the region of flux rope detection at the level of $R_{\mathrm{cutoff}} = 7\,R_{\mathrm{c}}$. This

flux rope is very curved, as evidenced by panel 5(b), as well as the pair of counter-rotating vortices of magnetic field lines at $y = -17\,R_{\mathrm{E}}$; $z = -6\,R_{\mathrm{E}}$ and $-7.5\,R_{\mathrm{E}}$ in panel 5(a). It passes over a string of four virtual spacecraft locations marked in panels 5(a)–5(b) and the magnetic field time series at these locations are shown in panels 5(c)–5(f). We do not proceed to a change of coordinate system to allow for direct comparison of the slices in panels 5(a)–5(b) with the virtual spacecraft time series. Due to the curved shape of the flux rope, the characteristic bipolar signature of the passing flux rope is not readily

observed in a single magnetic field component. However it is detected by the two lower virtual spacecraft (panels 5(e)–5(f)) during the time interval 1560–1580 s. They cross the leading portion of the flux rope, which despite its curvature has an axial orientation approximately along the $z$-direction, that is perpendicular to the tailward magnetosheath flow in the $-x$-direction. At the location of the lowest virtual spacecraft, the axis of the flux rope axis is nearly aligned with the $z$-direction so that the bipolar oscillations are well visible in the $B_x$ and $B_y$ components of panel 5(f). At the second location (panel 5(e)) the axis of

the flux rope is more inclined along the main diagonal of the $(x, y)$-plane so that the signature is less obvious but can still be inferred in the $B_y$ component. Both these virtual spacecraft also show a clear dip in the magnetic field magnitude $B$, indicative of their passing near the core of the flux rope where the field is close to zero, as confirmed by the sign flip of the out-of-plane components in panels 5(a) and 5(b). The two upper virtual spacecraft of panels 5(c) and 5(d) on the other hand are observing the trailing part of this flux rope, whose axis is much more aligned with the ambient flow direction, making it geometrically

impossible to observe a bipolar signature or the magnetic field magnitude approaching zero.

    Figure 6 tracks the propagation of a flux rope whose longitudinal axis is essentially parallel to the flow and the $x$-direction. Each pair of consecutive panels is in the same format as panels (a)–(b) of Figure 5, showcasing the three-dimensional structure of the flux rope. Its rolled-up magnetic field is clearly visible as a vortex of the black lines in panels 6(a), 6(c), 6(e), and 6(g) encompassed by the green contour, and the elongated structure parallel to the $x$-direction is clearly visible in panels 6(b),

6(d), 6(f), and 6(h) in both the colour plot and the green contour. A further guide to the location of this flux rope is given by the large-scale patterns in the out-of-plane $B_x$ in the left panels 6(a), 6(c), 6(e), and 6(g): the bipolar sign change of $B_x$ at $x \gtrsim -17\,R_{\mathrm{E}}$; $z \approx 0$ corresponds to the centre of the magnetotail current sheet separating the North and South magnetotail lobes. Furthermore, the transition from mostly southward field to less homogeneous orientations near $y = -20\,R_{\mathrm{E}}$ ($-18\,R_{\mathrm{E}}$ in panel 6(a)) corresponds to the magnetopause. Over the course of 300 s, this flux rope keeps its characteristic shape and orientation,

especially its longitudinal alignment with the magnetosheath flow and the $x$-direction. However it is seen to gradually shrink, from a diameter of over $2\,R_{\mathrm{E}}$ in panel 6(a) to only about $1\,R_{\mathrm{E}}$ in panel 6(g) in the end, and a length of over $15\,R_{\mathrm{E}}$ initially in



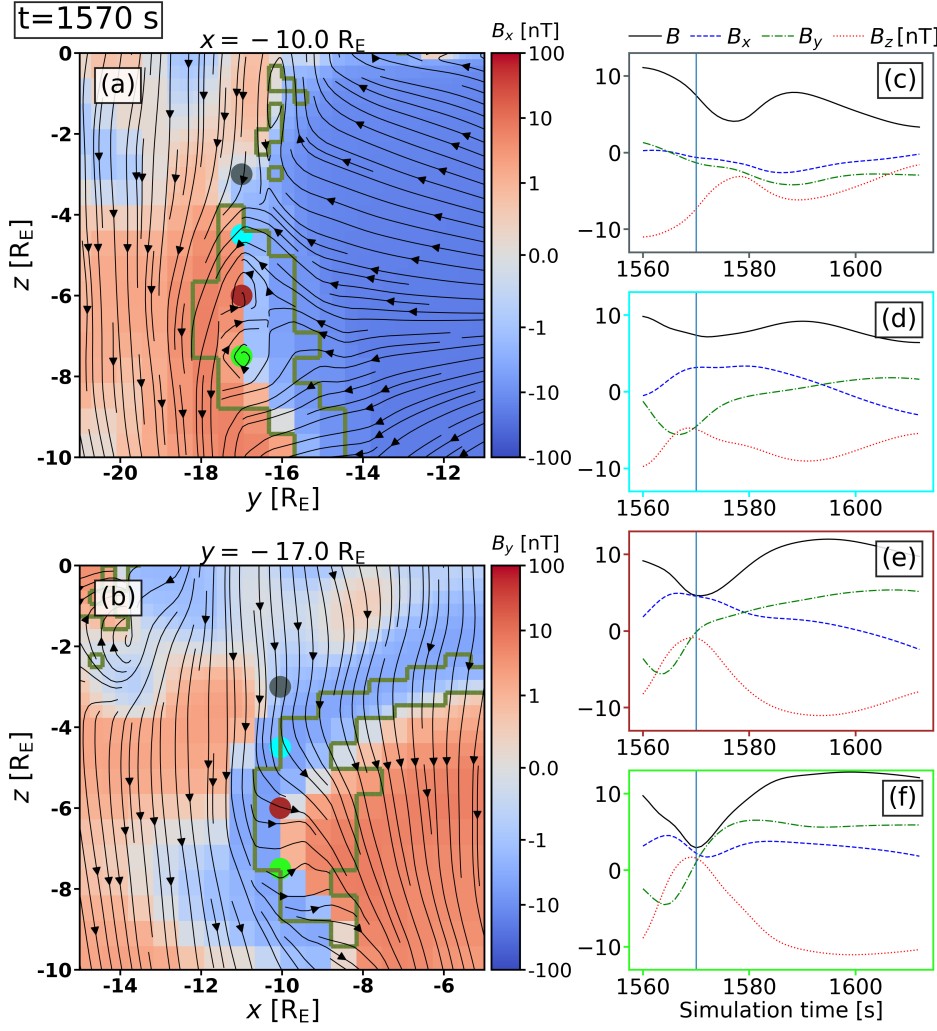

**Figure 5.** Panels (a) and (b) show a slice through a flank flux rope at time $t = 1570\,\mathrm{s}$ in the $(y, z)$-plane (resp. $(x, z)$-plane) at $x = -10\,R_\mathrm{E}$ (resp. $y = -17\,R_\mathrm{E}$). The colour shows the orthogonal $B_x$ (resp. $B_y$) component of the magnetic field whereas the black lines are tangent to the in-plane magnetic field. The green contour encompasses the regions where flux ropes are detected at the level of $R_\mathrm{cutoff} = 7\,R_\mathrm{c}$. Panels (c)–(f) show the time series of magnetic field magnitude and components for the time interval 1560–1612 s at the four virtual spacecraft locations marked in panels (a)–(b), with the vertical line marking the time of panels (a)–(b). See the animated version of this figure as Supplementary Material Animation 2.





panel 6(b) to about $5\,R_{\mathrm{E}}$ in panel 6(h). Crucially, at t=1600 s, what remains is a flux rope with a longitudinal axis that is parallel with the $x$-axis and a magnetic field configuration that has no shearing component with the southern lobe's predominantly $B_x < 0$ magnetic field which it is flowing against. Presumably any parts of the flux rope that were shearing with the lobes have

310 eroded through reconnection and are thus absent at later stages, such as the upstream, upwards-oriented end of the flux rope at $x > -12\,R_{\mathrm{E}}; z > 0$ which is shearing with the northern lobe's $B_x > 0$ in panel 6(b).

We provide a global overview of where flux ropes occur in the simulation domain in Figure 7. For the same level of $R_{\mathrm{cutoff}} = 7\,R_{\mathrm{c}}$ as has been used in Figures 4, 5, and 6, and for the time interval 1073–1612 s at a cadence of 1 s, the occurrence of flux ropes is recorded. This is plotted on a $y - z$ map with contours for consecutive ranges of the simulation domain along the

315 $x$-direction. As an additional guide, the cross-section of the tail lobes at $x = -10\,R_{\mathrm{E}}$ at $t =$1612 s is plotted in the background to provide an approximate location of the magnetopause encompassing the magnetotail lobes. The yellow contour for $x \in [4; 12]\,R_{\mathrm{E}}$ shows that all flux ropes on the subsolar dayside sunward of the cusps, which are therefore FTEs, occur within $|z| < 10\,R_{\mathrm{E}}$, with a small exception in the North near $y = 2\,R_{\mathrm{E}}$. This confirms what can be inferred from Supplementary Material Animation 1, which does not show any FTEs propagating significantly further than the cusps. The orange contour

for $x \in [0; 4]\,R_{\mathrm{E}}$ and the contours for all values of $x < 0$ show that no flux ropes occur at $|z| > 10\,R_{\mathrm{E}}$ in this simulation. The only excursion is seen near $y = -15\,R_{\mathrm{E}}; z = -11\,R_{\mathrm{E}}$ in the red and purple contours ($x \in [-70, 0]\,R_{\mathrm{E}}$) and it corresponds to the leading portion of the flux rope presented above in the $(-y, -z)$-quadrant of Figure 2(c), Figure 5, and Supplementary Material Animation 2. The deeper tailward, the lower in $|z|$ the flux ropes are located in the tail near the magnetopause. The range in $z$ is larger inside the magnetotail, corresponding to plasmoids formed by magnetotail reconnection. Through the influence

of various instabilities, the magnetotail current sheet is flapping, leading to significant deviations of its location from $z \approx 0$, as studied by Palmroth et al. (2023) and in further separate studies under preparation. As a consequence, tail plasmoids are detected in a wider range in $z$. The current sheet is nevertheless approximately centred on $z = 0$ in the absence of a geomagnetic dipole tilt or asymmetric driving conditions. The contour for $x \in [-110; -70]\,R_{\mathrm{E}}$ exhibits a wider spread in $y$ and $z$ which is due to the fact that in the early phase of the time interval considered, some large-scale magnetic structures, which originated in

the initialisation of the magnetosphere in this setup, are still in the process of being flushed out of the simulation domain. For the purpose of this study it was however deemed unnecessary to limit the box to a shorter range in $x$ or make that range vary with time.

## 5 Discussion

The results presented in Section 4 allow to track the motion of flux ropes from the dayside to the far tail, as discussed in the

335 following Section 5.1. They also give insight into the properties and limitations of the flux rope detection method presented in this work. The method is compared to the X- and O-line detection method by Alho et al. (2024) in Section 5.2, and its behaviour in higher guide field configurations is discussed in Section 5.3.



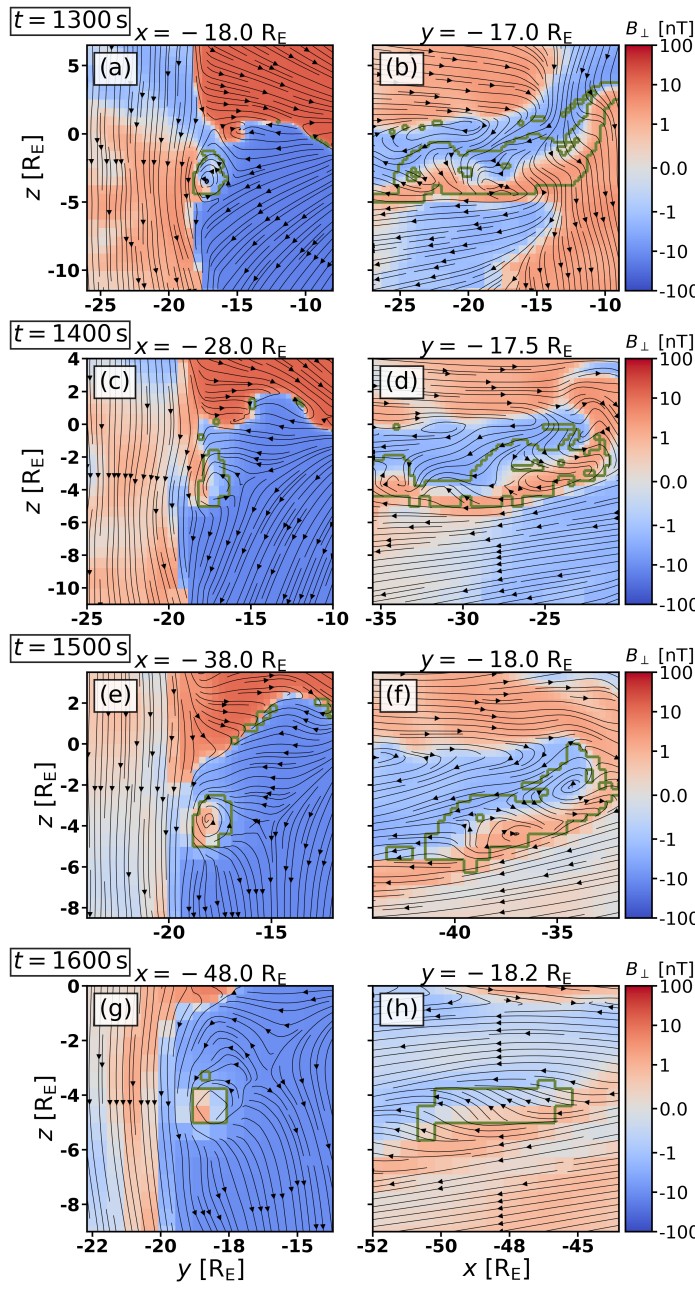

**Figure 6.** Left column panels (a), (c), (e), and (g) (right column panels (b), (d), (f), and (h), respectively) show slices through a flank flux rope in the $(y, z)$-plane (resp. $(x, z)$-plane) at times $t = 1300\,\text{s}$ (panels (a)–(b)), $t = 1400\,\text{s}$ (panels (c)–(d)), $t = 1500\,\text{s}$ (panels (e)–(f)), and $t = 1600\,\text{s}$ (panels (g)–(h)) tracking its tailward propagation. The colour shows the orthogonal $B_x$ (resp. $B_y$) component of the magnetic field whereas the black stream lines are tangent to the in-plane magnetic field. The green contour encompasses the regions where flux ropes are detected at the level of $R_{\text{cutoff}} = 7\,R_{\text{c}}$.

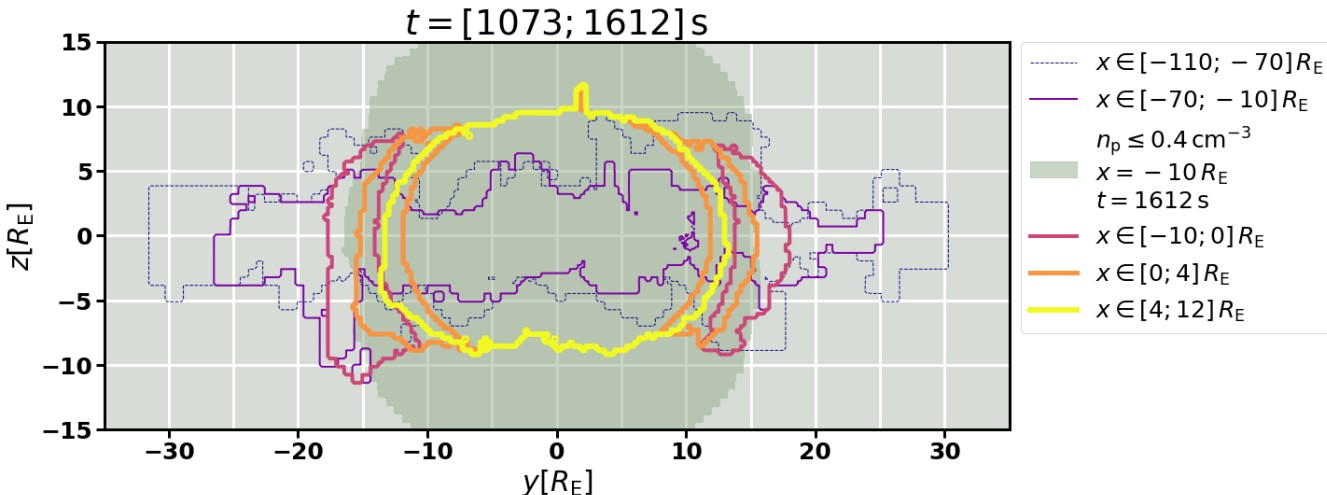

**Figure 7.** Map of flux rope detection at the level of $R_{\text{cutoff}} = 7\,R_c$, projected onto the $(y, z)$-plane. Each contour corresponds to a range of the simulation domain in the $x$-direction and delimits the region where a flux rope is detected at least once for the time interval 1073–1612 s at a cadence of 1 s. The darker shaded region denotes where the ion number density is lower than $0.4\,\text{cm}^{-3}$ at $x = -10\,R_E$ at $t = 1612\,\text{s}$, as a proxy for the location of the magnetopause and tail lobes.

## 5.1 Global evolution of flux ropes under southward IMF

In the simulation setup presented here (Section 3), namely under purely southward IMF and without geomagnetic dipole tilt,
flux ropes form due to magnetic reconnection at the dayside magnetopause near $z = 0$, as shown in Figures 2, 4, and 7 in Section 4. These flux ropes are advected by the local magnetosheath plasma flow, which is globally dominated by the hydrodynamic flow pattern of the shocked solar wind plasma around the magnetopause, but is also locally affected by magnetic reconnection exhausts (e.g., Hoilijoki et al., 2019; Pfau-Kempf et al., 2020). This means that flux ropes can flow into all quadrants of the $(y, z)$-plane, as shown in Figure 7.

At high $|z|$, these flux ropes are quickly eroding under the effect of magnetic reconnection, as seen in Figure 4 as well as Supplementary Material Animation 1. This is the natural consequence of the purely southward conditions, which produce flux ropes with little axial field and therefore an overall leading-edge field that is mostly antiparallel to the lobe field and reconnects efficiently with it. As a result, no such dayside-originating flux ropes survive past $x = 0$ at high $|z|$, as shown in Figure 7.

At lower $|z|$ and larger $|y|$, the flow advects these flux ropes further along the flanks of the magnetopause, as showcased in
Figures 3, 5 and Supplementary Material Animation 2, as well as Figure 6. When following the evolution of such flux ropes, it is notable that any section of the flux rope presenting a magnetic field configuration antiparallel to the lobe magnetic field will erode away due to magnetic reconnection. This can be seen as the disappearance of the trailing section of the flux rope tracked in Figure 6(b), which has reconnected with the antiparallel magnetic field of the northern magnetotail lobe. It is also evidenced by the gradual decrease in the $z$-range in which flux ropes occur further downstream, shown in Figure 7. An exception is the





flux rope of Figure 5, which is seen as low as $z = -11\,R_{\mathrm{E}}$. Its leading section is oriented such that its field is mostly parallel with the southern lobe magnetic field, so it is not expected to significantly erode unless it were to advect towards the northern lobe and then reconnect.

Flux ropes have been observed in the far tail at $x = -67\,R_{\mathrm{E}}$ (Eastwood et al., 2012), albeit not very often. One obvious reason for the low number of reported far tail magnetosheath flux rope observations is the paucity of measurements in that region of geospace due to the orbital configuration of most spacecraft missions. An exception is the Acceleration, Reconnection, Turbulence, and Electrodynamics of the Moon's Interaction with the Sun mission (ARTEMIS, Sibeck et al., 2011) used by Eastwood et al. (2012) to observe these far tail flux ropes. The conditions in the present work lead to the formation of flux ropes on the dayside with an axis mostly parallel to the $(x, y)$-plane that are prone to reconnecting rapidly with the lobe magnetic field. The surviving ones such as the flux rope in Figure 6 would be extremely difficult to observe due to their axis being parallel to the plasma flow in the $-x$-direction, precluding the observation of the bipolar signature of the transverse magnetic field. It is therefore possible that such $x$-aligned magntosheath flux ropes are common, but difficult to detect. In conditions with significant IMF $B_y$, flux ropes may form in orientations more favourable for subsequent observations, such as in the event of Eastwood et al. (2012). In that event, it could be speculated that the lower portion of the flux rope in their Figure 7 might have eroded against the southern lobe field and that the flux rope was better preserved at the location of the ARTEMIS P1 spacecraft near the tail current sheet. However the authors also refer to conditions such as the fast shear flow between the magnetosheath and the lobe in the far tail region, which can suppress reconnection and help in preserving flux ropes in the tail (La Belle-Hamer et al., 1995; Cassak, 2011).

It will be the subject of future investigations using this novel flux rope detection method to study the global evolution of flux ropes under different IMF conditions. Once such a set of simulation datasets presenting a variety of geomagnetic and upstream conditions is available, it may become feasible to provide quantitative predictions on the occurrence rate of such low-latitude, flow-aligned, far-tail flux ropes, yet such a prediction would be perilous on the basis of the present simulation only. Another approach that may be considered, and would require a separate analysis, is to follow the approach developed by Grandin et al. (2024); Guo et al. (2024) who quantified the potential signature of dayside FTEs in soft X-ray observations. The Solar wind Magnetosphere Ionosphere Link Explorer (SMILE Branduardi-Raymont et al., 2018) and Lunar Environment heliospheric X-ray Imager (LEXI Walsh et al., 2024) missions will produce soft X-ray images from the emissions generated by charge-exchange reactions in near-Earth space (see Sibeck et al., 2018, for a review on the technique). SMILE will have a vantage point from "above the poles" with its eccentric, highly inclined orbit, while LEXI will observe from the surface of the Moon. Both will thus provide complementary and unprecedented observations, which may help in observing flux ropes in and around the magnetosphere more comprehensively than has been possible hitherto.

## 5.2 Flux ropes and O-line topology

In parallel to the generic flux rope detection method presented in this work, Alho et al. (2024) developed a local method determining the spatial location of lines where the magnetic field is in the so-called X- and O-line configurations, using a combination of the MGA and MDD techniques. X-lines are the site of magnetic reconnection when a non-zero rate of energy





conversion and field topology change is observed. O-lines can be interpreted as the core of magnetic flux ropes. As the method
uses derivatives of the magnetic field in a local coordinate system, it yields the essentially one-dimensional axis of flux ropes.
In order to identify a flux rope based on the O-lines yielded by that method, field lines could be traced in the neighbourhood
of said O-lines, for example. In the case of larger flux ropes where the axial region of 'straight' field lines is wider than a
one-dimensional line, the Alho et al. (2024) method might not yield a clear O-line. On the other hand, the method described
in the present work does not necessarily detect the actual core of flux ropes. In the case of a perfectly zero axial field, the
tracing would likely not succeed on a discrete grid, whereas in the case of non-zero axial field $R_c$ could become very large and
tracing proceed to overly large distances. The method will however detect neighbouring regions surrounding the core reliably,
as shown in this work.

Although as explained in Section 2 the detection method is designed to be independent of specific spatial scales, one never-
theless has to set the $L_{\mathrm{max}}$ parameter, which defines the maximum distance tracing is allowed to proceed forward and backward
along the field line. Of course $R_{\mathrm{max}}$, the maximum extent away from the staring point allowed for flux rope detection, is a
parameter too but it cannot be larger than $L_{\mathrm{max}}$ and should not be set too small so that the optimal $R_{\mathrm{cutoff}}$ can be deter-
mined for the subsequent studies at hand. By setting the $L_{\mathrm{max}}$ parameter to $12\,R_c \approx 4\pi\,R_c$, an a priori decision is made to
search for well-formed flux ropes with the magnetic field clearly winding around their axis. This naturally includes FTEs at
the magnetopause and magnetotail plasmoids (see Section 2). Setting lower values for $L_{\mathrm{max}}$ increases the likelihood of false
positive identifications, namely regions of generally bent or curved magnetic field such as current sheets, which are however
not necessarily winding into flux rope structures. Additionally, at the chosen level of $R_{\mathrm{cutoff}} = 7\,R_c$, it is visible in Figure 4
and Supplementary Material Animation 1 that all detected flux rope regions (green circles) include an O-line (yellow squares).
However, a number of O-lines are seen outside of detected flux rope regions. While there might be positive flux rope detections
just outside the plane of the respective panel, it is more likely that such isolated O-lines exhibit the O-line topology locally
but not at a sufficiently large scale with respect to the grid resolution to be picked up by the tracing method of this work with
$L_{\mathrm{max}} = 12\,R_c \approx 4\pi\,R_c$ and $R_{\mathrm{cutoff}} = 7\,R_c$.

These two methods are therefore complementary in their approach and results. The local method of Alho et al. (2024) detects
all O-line configurations regardless of the wider surrounding magnetic field configuration, whereas this work demonstrates a
field-tracing method geared towards identifying well-formed flux ropes of any sufficiently-resolved scale.

## 5.3 Higher guide field configurations

In this work, flux ropes are identified in a simulation setup featuring purely southward IMF and no dipole tilt. These conditions
lead to magnetopause reconnection with (nearly) zero guide field and therefore the formation of flux ropes with low axial
field. When introducing an IMF $B_y$ component, magnetopause reconnection will occur in different locations (e.g., Trattner
et al., 2021), leading to a different sectorial distribution of the flux ropes but also potentially more complex topologies of
flux ropes (Fargette et al., 2020). This will also have an impact on the distribution of flux ropes further down the flanks and
their reconnection with the lobes or their survival down the tail. Another aspect of the introduction of a non-zero $B_y$ is that
this introduces a guide field to the reconnection geometry. In (mostly) antiparallel reconnection as in this work, flux ropes



consist of magnetic field that is close to perpendicular to the flux rope axis. With a stronger guide field, the flux ropes will be less tightly-wound. The magnetic field at the core of such flux ropes is then oriented more parallel to the flux rope axis. This

case is less favourable to detection by the present method. However the magnetic field is still wrapping around and thus more perpendicular to the axis of the flux rope in outer layers, allowing detection albeit potentially requiring a higher $R_{\mathrm{cutoff}}$.

It is possible that a higher $R_{\mathrm{cutoff}}$ is needed to identify all flux ropes in such higher guide field configurations. Even though the method presented in this work might not be able to detect all cells at the core of a flux rope with stronger axial field, it should nevertheless detect the parts of the structure surrounding the core, allowing identification of the flux rope.

**6  Conclusions**

We present a new method to detect flux ropes at runtime in large-scale numerical simulations of the Earth's magnetosphere. The method is implemented in the hybrid-Vlasov model Vlasiator. Using only magnetic field line tracing, the method detects flux ropes of any scale and orientation as structures where the magnetic field is sufficiently wound up. The method does not require a priori identification of, for instance, the magnetopause or the tail current sheet or bipolar magnetic field profiles. It

is thus more general and robust than previously published flux rope detection methods, and may find applications beyond the specific implementation used in this study. The key aspect of the method is to scale the search criteria by the local curvature radius of the magnetic field, which enables an identification of flux ropes that is agnostic to their absolute spatial scale.

We apply the flux rope detection method to a simulation of the magnetosphere under purely southward IMF, which produces abundant FTEs generated by dayside magnetic reconnection as well as magnetotail plasmoids generated by tail current sheet

reconnection. We then analyse in particular the global evolution of FTE flux ropes along the magnetopause, which is characterised by rapid erosion due to reconnection with the shearing magnetic field component of the cusp and lobe magnetic field. This leads to the consequence that no FTEs survive for more than a few Earth radii tailward from the cusp regions. Lower on the flanks however, FTE flux ropes without shear component are advected with the magnetosheath flow into the far tail and preserve their integrity for tens of Earth radii.

Future research will include the analysis of the formation and evolution of tail plasmoids, which is not addressed in this work. Furthermore, building upon the flux rope detection capabilities demonstrated here, following studies will be able to quantitatively study the evolution of flux ropes as pioneered in earlier simulation and observational works (e.g. Akhavan-Tafti et al., 2019; Hoilijoki et al., 2019; Paul et al., 2023). Simulations with different upstream conditions will shed more light onto the global behaviour of FTEs, potentially allowing the investigation of the interplay of flux ropes and the Kelvin-Helmholtz

instability at the magnetopause, or the suppression of lobe reconnection under different magnetospheric or upstream conditions. Finally, beyond the arguably difficult in situ observation of far-tail flux ropes, it also remains to be quantified whether soft X-ray imagers like SMILE and LEXI will be able to observe them, as has been suggested for the observation of dayside FTEs with SMILE (Grandin et al., 2024; Guo et al., 2024).



*Code and data availability.* Vlasiator (Pfau-Kempf et al., 2024) is open-source under the GNU GPL-2 license and hosted at GitHub (https: 455 //github.com/fmihpc/vlasiator). The dataset used and presented in this work requires multiple terabytes of storage and dedicated infrastructure for its handling and analysis. It can be shared upon reasonable request to the authors.

*Video supplement.* The animated version of Figure 4 is presented in Supplementary Material Animation 1. The animated version of Figure 5 is presented in Supplementary Material Animation 2.

*Author contributions.* Conceptualisation: YP, MA, KH; Data curation: YP, JS, MA; Formal analysis: YP, MA; Funding acquisition: YP, 460 MP; Investigation: YP, KH; Methodology: YP, MA, KH; Project administration: YP, MP; Resources: YP, MP, MB, UG; Software: YP, UG, KP, MB, MA, LP; Supervision: YP, MP; Verification and validation: YP; Visualisation: YP, LP; Writing — original draft preparation: YP; Writing — review and editing: all co-authors.

*Competing interests.* Some authors are members of the editorial board of Annales Geophysicae.

*Acknowledgements.* YP wishes to thank the teams around T. Pulkkinen and M. Liemohn at CLASP, University of Michigan for fruitful 465 discussions during his visit. Ivan Zaitsev is also acknowledged for constructive discussions and suggestions during the writing process.

YP acknowledges the Research Council of Finland grant number 339756-KIMCHI. KP acknowledges the Research Council of Finland grant number 336805. GC, KH, MA, MP, LP acknowledge the Research Council of Finland grant numbers 347795, 345701, 352846, and 361901. MB and UG acknowledge EuroHPC "Plasma-PEPSC" Centre of Excellence (Grant number 4100455) and the Research Council of Finland matching funding (grant number 359806). MG acknowledges the Research Council of Finland grants 338629-AERGELC'H and 470 3604333-ANAON. The work of MA is also funded by the European Union (ERC grant WAVESTORMS - 101124500). Views and opinions expressed are however those of the author(s) only and do not necessarily reflect those of the European Union or the European Research Council Executive Agency. Neither the European Union nor the granting authority can be held responsible for them.

The simulation presented in this work was run on the LUMI-C supercomputer through the EuroHPC project Magnetosphere-Ionosphere Coupling in Kinetic 6D (MICK, project number EHPC-REG-2022R02-238). Analysis was performed on the "Carrington" Cluster of the 475 University of Helsinki. We wish to thank the Finnish Grid and Cloud Infrastructure (FGCI) for supporting this project with computational and data storage resources. We used the VisIt (Childs et al., 2012), Analysator (Battarbee et al., 2024), and yt (Turk et al., 2011) tools to produce the Figures and the Supplementary Material Animations.



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
