# Peer review of "Global evolution of flux transfer events along the magnetopause from the dayside to the far tail"

_Annales Geophysicae, 2024_

## Author Comment (AC1)

**Response to Referee Comment 1**

**Yann Pfau-Kempf on behalf of all co-authors**

**March 28, 2025**

We thank the referee for the kind words and constructive review which will help us improve the paper. We reply point-by-point below (*in italics*).

Line 41: The authors use the term flux rope for both FTE and plasmoid, but what about Kelvin Helmholtz vortices? They are not mentioned until the very last paragraph in the conclusions. If they are, or are not included in the 'flux rope' definition here it should be mentioned.
*We will clarify that flux ropes can form as part of the development of the Kelvin-Helmholtz instability, as observed e.g. by Hwang et al. (2020)* https://doi.org/10.1029/2019JA027665.

Figure1: Have you looked at the boundary points in terms of the points which fail only one of either R+ or R-? They would be the edges of the flux rope structure and might be regions of interest themselves.
*These points will certainly be of interest in more focussed studies on particular flux ropes, but they would clutter the figures in this work too much. Any point along the field line would be a one-sided detection, thus flagging significantly larger volumes than just the vicinity of the flux rope.*

Line 115: Why 'significantly larger than the domain size'? I would assume if the traced point leaves the domain then surely the point is not flux rope.
*As explained in Section 2.3, due to the discretisation it is possible some traces keep circling around a certain region for long distances, hence the need for such termination conditions.*

Line 145: How is the fraction of omitted cells decided? Is it random, or geometric conditions set by the user?
*This is left to the appreciation of the user. In our runs, at most 0.05% of cells were left unresolved for the full-domain tracing yet it was ensured all flux-rope detections had completed. This reduced the time spent on the tracing algorithm by an estimated 30–50%. We will add a mention of this in the text.*

Line 165: What radius are the field aligned currents coupled from? The data availability statement notes the large size of the output data, but it may beneficial to upload a simulation parameter input file ( or text file summarizing

the simulation input parameters) for future comparisons.

*The coupling radius is set to 5.6 $R_{\mathrm{E}}$ in this run, we will add this information to the text. The dataset including the configuration files is now publicly available and we will include the reference to it.*

http://urn.fi/urn:nbn:fi:att:3ce0f038-2c69-4c7c-8f67-7a71e9e57b56

Figure 2: This is where a quantified measure could be helpful. In the snapshot it's clear there are more points identified with a larger Rcutoff, and at least one flux rope is missing between 3Rc and 7Rc. What I would like to see is what is the total flux rope volume for each of these panels? This could then be included in the video or even shown as a time series with a curve for each Rc setting. It could help justify the selection of Rcutoff = 7Rc.

*This is a good suggestion. We propose to include the figure below and discuss it. It shows the total volume of the detected simulation cells as a function of time for various $R_{\mathrm{cutoff}}$ in panel (a) and as a function of $R_{\mathrm{cutoff}}$ for various times. These show that there is a range of suitable values around 7. A much lower cutoff value (below 5–6) leads to significantly less volume detected, whereas much higher values (beyond 8) lead to a rapid increase in the detected volume.*

[Figure]

Figure 4: The + and circles in panels e and f are difficult to distinguish between.
*We will change the markers used to distinguish them better.*

Line 245: With both the yellow circle O points and the green flux rope points, perhaps the coverage of the O points (within some distance threshold) could be reported here and used as validation. It could be reported as a percentage over time, again for the different choices of Rcutoff to show that 7Rc performs the best.

*Without an adaptation of the distance threshold around O lines due to varying scales, we think this would be too crude an approach. On the other hand, a more rigorous comparison would in the end require some determination of the volume surrounding either O lines or detected cells*

*(this study's method), by tracing or other proxies (like Paul et al., 2022), which we would rather leave for upcoming studies. We think the spatio-temporal correspondence of the yellow circle O markers and the regions detected in green is sufficient as a cross-validation at this stage.*

Figure 5: The satellite traces may show a better structure crossing if put into LMN coordinates.
*The "traditional" LMN coordinate system based on the normal direction to the magnetopause is difficult to determine in this case as the magnetopause has a rather complicated shape. However performing a Minimum Variance Analysis in order to single out the component with least variation allows to extract the flux rope's signature more clearly and we will include this in the figure.*

Figure 6: I believe the X and Y scaling (horizontal axes) is different between some panels in the same column. This made it difficult to understand the point about the cross section shrinking. It would improve comparison between panels in a single column to have the scales the same.
*The frames are scaled to best highlight the flux rope structure, especially with the magnetic field stream lines. It would be harder to make out with the larger frame of the top panels for the lower ones. We will add a reference scale on the plots to guide the eye and better illustrate the varying frame sizes.*

Figure 7: Great use of perspective to summarize the flux rope detection results. Again, could be improved with an indication of how many flux ropes were found for each contour level. If identifying contiguous flux ropes is not currently possible, at least the total number of points (or volume of all points?).
*We have not yet developed a method to identify and track single flux ropes. However, we think it would make the figure difficult to construct and understand when adding the information of the coverage volume that would sum over spatial dimensions as well as time.*

Line 349: Figure 7 demonstrates that no dayside flux ropes survive to X=0, but is there a gradient or a sudden cutoff? If it is reconnection eroding the structure, maybe larger FTEs make it further downstream before vanishing?
*As can be seen in Figure 4 and especially the Supplementary Material Animation 1 indeed larger FTEs survive a little longer along the magnetopause, so it is not completely abrupt, but given the steep inclination of the magnetopause it is indeed so that all FTEs in the noon sector erode away between 4 and 0 $R_{\mathrm{E}}$.*

Line 351: "any section of the flux rope presenting a magnetic field configuration anti-parallel to the lobe magnetic field will erode away due to magnetic reconnection." Consider rewording slightly. The results only show a single flux rope that has its anti-parallel portion eroded, while every flux rope with such conditions could experience such erosion, it was not shown that

this occurs in every instance. If the results do show this, then that should be included explicitly in the results section.

*We are led to this interpretation through the combined evidence of the FTEs eroding through reconnection at high-|z| in Figure 4/Supplementary Material Animation 1 and Figure 7, the case study shown in Figure 6, and other flank flux ropes we have investigated but not shown here. We will reword this paragraph to better reflect this.*

Line 356: Can it be estimated from these results what percentage of flux ropes are vanishing over the poles vs surviving downstream? Does it match some geometric ratio of the portion of the dayside X line length?
*See next response.*

Line 377: Similar to above, it's mentioned that 'such a prediction would be perilous'. While I agree that a single simulation should not be over-extrapolated, the results can certainly be reported. How many low latitude flux ropes were found for this time interval? How does that compare to the number that vanished over the poles? The answer will be limited to this simulation setup, but nonetheless interesting.

*In a global-scale, fluid-like picture a simplified geometric argument could be likely made. However e.g. Figure 2, and our previous work (Pfau-Kempf et al., 2020, $https://doi.org/10.1063/5.0020685$), show that the geometry of the reconnection site is non-trivial and leads to intricate structures such as the curved flux rope from Figure 5/Supplementary Material Animation 2. Once a method is developed to characterise flux ropes as distinct objects, it will be easier to try and address this issue by investigating the shape and orientation of the flux ropes and their evolution as a function of time and magnetopause clock angle.*

*We however suggest to provide an estimate of the rates of occurrence based on the figures and animations of this work. From Supplementary Material Animation 1 one can estimate that 6 (5) FTEs vanish over the North (South) cusp over the interval of 539 s, yielding an occurrence rate of about 1 FTE per 100 s per hemisphere. In Figure 3, one can identify 4–5 flux ropes between 0 and $-100\,R_{\mathrm{E}}$ on each flank, while Figure 6 provides an estimate of the transport velocity, namely $30\,R_{\mathrm{E}}$ in 300 s, or $100\,R_{\mathrm{E}}$ in 1000 s, which means that there is one flank flux rope passing by in about 200–250 s on each flank. This compares remarkably well with the observations of Eastwood et al. (2012) so we will include this estimate in the discussion.*

Line 397: Does this imply that there may be holes that form in the field of flux rope points? Will this then make it more difficult to identify individual contiguous flux rope structures?
*Indeed, theoretically the curvature radius is very large at the centre of a flux rope with axial field and the method may not flag that region as sufficiently rolled up. In practice, with the discretisation of the model setup taken into account, we do not expect this to lead to significant holes. Refining*

*the identification of contiguous structures with local proxies should yield robust results. We will clarify the discussion in this respect.*

Line 449: How would this method tell the difference between a rolled up KH vortex and a low latitude flux rope which has been carried downstream?
*This tracing method by itself does not characterise flux ropes further. But it serves as a starting point to identify individual flux ropes as distinct objects, as mentioned above. Further information such as the forward or backward end points of the field lines could be used to discriminate between the low-latitude flux ropes and KH vortices, or heat flux signatures (e.g. Tarvus et al., 2023, https://doi.org/10.3847/1538-4357/ad697a). We will add these considerations to that paragraph.*

---

## Author Comment (AC2)

**Response to Referee Comment 2**

**Yann Pfau-Kempf on behalf of all co-authors**

**March 28, 2025**

Thank you very much for the kind words and constructive review, which will help improving this paper. We reply point-by-point below (*in italics*).

Line 1: The term "rolled-up" is used to describe magnetic flux ropes. While not incorrect, "helical magnetic field" is more commonly used and precise. Also, the term "longitudinal axis" is not clear and should be clarified as it can vary in different space plasma contexts.
*We will rephrase this sentence.*

Line 16: The word "twisting" is used to describe the magnetic field geometry of flux ropes. Again, "helical" is a more accurate and commonly used term.
*We will also rephrase this sentence.*

Lines 18-21: The description of flux rope formation suggests they originate inside the Sun and pass through the solar surface. I do not think this is the case. References are needed if this is the case. On the other hand, this should include the possibility of formation by magnetic reconnection in the solar wind, as noted by Cartwright and Moldwin (2008) and Feng (2010).
*The emergence of flux ropes is the topic of both references that were given. We will add the possible formation mechanism through reconnection in the IMF and the references, thank you for this suggestion.*

Line 25: The phrase "When the magnetotail current sheet disrupts and reconnects, flux ropes form" should be revised to "When the magnetic field reconnects in the magnetotail current sheet, flux ropes can form."
*It will be revised.*

Lines 52-53: May mention the automated method developed by Li et al. (2023) for detecting FTEs in Mercury's magnetosphere.
*Thanks for pointing out this method too! It will be included.*

Figure 1: Consider including a panel showing the curvature radius along with the magnetic field line for better illustration.
*We will include this information in the figure.*

Section 2.2. The simulation in this study was under a pure southward IMF. Therefore, the flux ropes resulted from this simulation likely do not have strong core

field. How does the core field influence the criteria set up here?
*The effect of a stronger core field is discussed in Section 5.3. We will rephrase the section slightly to distinguish reconnection guide field from the flux rope core field, which were both called guide field so far.*

On the other hand, is it possible to investigate the curvature radius in a flux rope event with a strong core field or a flux rope without core field in the simulation? As shown in Sun et al. (2019, 2019GL083301) and Smith et al. (2024), flux rope with strong core field corresponded to a maximum in curvature radius, while without strong core field a minimum in curvature radius.
*The method certainly remains applicable in higher core field configurations. Although it will possibly not detect points near the core of the flux rope, it will still detect surrounding regions and thus allow identification of the flux rope. This is also discussed in Section 5.3.*

Lastly, flux transfer events often correspond to coalescence, i.e., merging of neighboring flux transfer events, as shown in Sun et al. (2022, angeo-40-217-2022 and many other simulation and observation works). Is it possible for this technique to identify those events?
*Future work based on the detected flux rope regions will certainly allow identifying individual flux ropes, tracking them and why not seeing if they coalesce with each other.*

Line 162: Clarify the term "Neumann (copy)."
*We will rephrase to explain the Neumann condition, which is the same as copy condition.*

Line 168: Revise "for on" to "on."
*We will rephrase this.*

Line 205: I think that I can identify the "long flux rope" and the "curved flux rope". However, it would be better to identify them in the Figure.
*We will highlight in Fig. 2 and 3 the flux ropes in question as well as the flux ropes that are shown in Fig. 5 and 6 to make it easier to follow.*

Line 226: Add citations for MGA and MDD.
*We will add the reference to the review by Shi et al. (2019) https://doi.org/10.1007/s11214-019-0601-2.*

Figure Captions (Figures 5 and 6): Clarify the abbreviation "resp."
*Will be clarified.*

Figures 5c to 5f, may include horizontal lines at y = 0.
*This will be included.*

Figure 6 pretty nicely shows the flux ropes!
*Thank you!*

Lines 280 to 281: Is it possible that the counter-rotating vortices of magnetic field lines are coalescing flux ropes as I mentioned earlier?

*As they are counter-rotating the field is parallel and not antiparallel in between, as shown by the in-plane magnetic field lines in Figure 5(a). So there cannot be any coalescing through magnetic reconnection here. Taking the red-to-blue transition as being close to the flux rope core, Figure 5(a) and (b) show that the flux rope is bent in a crescent shape.*

Line 290: "more inclined along the main diagonal of the (x,y)-plane" Is this place trying to say that the axis of the flux rope is mainly in the x-z plane?

*Yes indeed. The paragraph was rephrased and expanded to better convey the shape of the flux rope.*

In Figure 7, the contour for the x within 4 to 12 RE is larger than the contour for the x within 0 to 4 RE. I could not understand why. Could the authors explain more about this?

*The larger, yellow contour from 4 to 12 $R_E$ comprises essentially all of the subsolar dayside magnetopause including the region where the FTEs reconnect with the cusp/lobe magnetic field. The next, orange contour at 0 to 4 $R_E$ only contains flux ropes that are lower in z and flowing towards the flanks.*

Line 341: Consider using "convected" instead of "advected." Same for other places.

*This will be rephrased.*

Line 390: Clarify the term "one-dimensional axis."

*The axis is a one-dimensional line, as opposed to the three-dimensional volume of a flux rope. This will be rephrased to be clearer.*

Line 392: Explain the use of "said."

*This will be rephrased.*

Line 434: Revise "priori" to "prior" or "previously."

*This will be revised.*

Line 437: Clarify the term "agnostic."

*This will be rephrased.*

Line 441: Consider using "antiparallel" instead of "shearing."

*This will be corrected.*

Line 443: Revise "shear component" to "antiparallel component."

*This will be revised.*

Line 444: Consider using "property" instead of "integrity."

*This will be changed.*

Line 447: I think that with Vlasiator simulations, it is also possible to investigate the energizations of protons and electrons as well as the important of flux transfer events in transferring magnetic flux and particles in the space

plasma physics (Section 2.1 in Sun et al., 2012, https://doi.org/10.1007/s11430-021-9828-0).

*We fully agree and this will be added.*